# Sensor-supported measurement of adaptability of dogs (*Canis familiaris*) to a shelter environment: Nocturnal activity and behavior

**Janneke E. van der Laan** \* , **Claudia M. Vinke** , **Saskia S. Arndt**

Faculty of Veterinary Medicine, Department of Population Health Sciences, Division of Animals in Science and Society, Animal Behaviour Group, Utrecht University, Utrecht, The Netherlands

\* J.E.vanderLaan@uu.nl

**Data Availability Statement:** The datafile is available from the Open Science Framework database (DOI: OSF.IO/J5XK2).

## Abstract

Every shelter dog is faced with the challenge to adapt to a kennel environment. To monitor the welfare of individual shelter dogs, evaluating behavioural and physiological parameters, potentially useful as indicators for adaptability of individual dogs is crucial. Nocturnal activity, i.e. resting patterns, has already been identified as a candidate indicator of adaptability and can be easily measured remotely with the help of sensors. We investigated the usefulness of a 3-axial accelerometer (Actigraph®) to monitor nocturnal activity in shelter dogs every night during the full first two weeks in-shelter starting directly at shelter intake, as a measure of welfare. Additionally, urinary cortisol/creatinine ratio (UCCR), body weight and behaviour data were collected to evaluate stress responses. A control group of pet dogs in homes, matched to the shelter dog group, was also monitored. Shelter dogs had higher nocturnal activity and UCCRs than pet dogs, especially during the first days in the shelter. Nocturnal activity, both accelerometer measures and activity behaviour, and UCCRs decreased over nights in the shelter. Smaller dogs had higher nocturnal activity and UCCRs than larger dogs and showed less autogrooming during the first nights. Dogs with no previous kennel experience had higher nocturnal activity and UCCRs, and showed less body shaking, than dogs with previous kennel experience. Overall, sheltered dogs also showed less body shaking during the first night. The number of dogs showing paw lifting decreased over days. Age class and sex effected only few activity behaviours. Shelter dogs significantly lost body weight after 12 days in the shelter compared to the moment of intake. Shelter dogs had disrupted nocturnal resting patterns and UCCRs compared to pet dogs and seem to partly adapt to the shelter environment after two weeks. Sensor-supported identification of nocturnal activity can be a useful additional tool for welfare assessments in animal shelters.

## Introduction

An animal shelter environment comprises many potential stressors for dogs (*Canis familiaris*), such as high noise levels, separation from attachment figures and the presence of unfamiliar

**Funding:** This work was supported by Mars Incorporated, through the Utrecht University Fund, by funding accelerometer equipment, camera systems, data storage and the analysis of urinary samples. The grant was awarded to the project, and therefore to all authors. The funders had no role in study design, data collection and analysis, decision to publish, or preparation of the manuscript.

**Competing interests:** The authors have declared that no competing interests exist.

dogs or people [1–3]. This poses a risk for welfare if dogs fail to adapt to these stressors [4,5]. Individual stress responses can be variable [6]. Therefore, monitoring an individual's adaptive response is key to understand and manage stress in a shelter.

When monitoring a stress response, preferably, both physiology (e.g. autonomous nervous system and hypothalamic pituitary adrenal axis parameters such as glucocorticoids) and behaviour of the individual are commonly evaluated [7,8] for sound conclusions. For instance, the glucocorticoid cortisol, which is widely used in many species to evaluate stress, has been found to be elevated in dogs for days up to weeks after they entered a kennel environment [3,9–11]. The dynamics of the cortisol response, however, can depend on factors such as previous experience of the dogs [8,12] and the size of the dog [13]. Addressing behaviour, dogs in kennels showed significantly less time lying down and resting, and significantly more time being alert, sitting, standing, travelling, and panting, compared to home-housed dogs [9], which may be connected to daily routines, but the behaviour of dogs is also influenced by social and spatial restriction during kennelling [14]. Over time, behavioural responses can change when dogs habituate to the kennel environment, which can for example be reflected by an increase in drinking and grooming, and a decrease in both panting and paw-lifting [12] and a decrease in behaviours associated with fear [15]. This is especially meaningful in connection to changes in cortisol.

However, cortisol responses and behaviour, among other welfare indicators, are not easy to measure in a practical setting as these methods are costly or time consuming. Sensor-based measurement of welfare-related parameters can be a useful addition to animal welfare assessments in practical settings, as sensors are becoming cheaper and can be used remotely and continuously [16]. One candidate parameter for sensor detection is activity- and resting behaviour of the animal, whereby the disturbance and recovery of resting behaviour can be an interesting variable to evaluate adaptability to a changing and/or stressful environment (e.g. in rats and farm animals [17–19]). Also, dogs in shelters may be subjected to disturbed rest, due to for example unfamiliar noises in the shelter [3]. Changes in resting patterns are therefore a parameter of interest in assessing the welfare of sheltered dogs. Accelerometers can be used to measure acceleration and therefore also physical activity, and provide information on frequency, duration, and intensity of (in)activity [13,20,21].

Previous studies have shown that dogs in an animal shelter can have aberrant activity and resting patterns, which especially manifests during the first period in the shelter and shows relations to other welfare measures. For example, Owczarczak-Garstecka & Burman [22] showed that shelter dogs that spent more time resting during the day had a more positive outcome in a cognitive bias test, showed less repetitive behaviour in the kennel and were more often coded as 'relaxed' by staff in the shelter. Also, time spent sleeping increased from 0% up to 42,7% in elderly dogs (8–12 years) during the first 6 days (between 14:00–16:00) in a shelter [23]. Adams and Johnson [24] observed a decrease in nocturnal sleep-wake episodes in one (n = 1) institution-housed dog from night 1 to night 2 and 3 after the dog entered this confined institution environment. They also observed no 'active sleep' (REM sleep) during the first night in this dog, which compares to a 'first night effect' in humans [25]. Accelerometer-based studies showed that sheltered dogs were more active than dogs in homes during three quarters of the day, including the night [26], and that activity was positively correlated to salivary and urinary cortisol levels [27]. Next to that, in one of our earlier studies we found nocturnal activity measured with accelerometers to be higher during the first few nights in the shelter but to decrease from night 1 and 2 to night 12 in the shelter [13]. However, it is still unclear how the dynamics of this nocturnal (individual) activity response change during these first twelve days in the shelter and how they relate to behavioural responses to stress.

Therefore, we investigated the usefulness of sensor-based assessment of nocturnal activity patterns as a measure of welfare in shelter dogs, by collecting information over time during the first 13 days in the shelter. We monitored nocturnal resting patterns using accelerometers, the ActiGraph®, and nocturnal behaviour by video observations, and additionally combined parameters of urinary cortisol/creatinine ratio (UCCR) and weight (loss) controlled for body condition score (BCS), during the first twelve days after entree in the shelter and compared their responses to a control group of pet dogs.

## Materials and methods

### Subjects

**Shelter dogs.**   Fifty-five dogs entering the largest animal shelter in the Netherlands (Animal Shelter DOA) between October 2018 and September 2019 were included in the study. Dogs were included in our study if they were between 1 and 13 years old, healthy and suitable to be handled (e.g. not too anxious or aggressive to approach, based on evaluation by the caretakers and the researchers). Five dogs of the included dogs were returned to their owners after 5–7 days, those dogs were therefore monitored shorter than the full 13 days. We included 20 female (10 intact, 7 neutered, 3 unknown) and 35 male dogs (23 intact, 12 neutered, see S1 Table for demographics of all dogs). Individuals were of various breeds and ages (mean = 3.7 years, range 1–13 years). Dogs were either strays (n = 17), relinquished by their owners (n = 34) or being temporary sheltered in crisis situations, when an owner had to be hospitalized or when taken for other reasons into custody (n = 4). Dogs were assigned to different weight classes (body sizes: <10kg, n = 19; 10-20kg, n = 13; 20-30kg, n = 12; >30kg, n = 11 [28]). For most dogs (n = 38), it was not known whether they had stayed in a kennel environment before, but it was known that 9 dogs stayed in a kennel before and 8 dogs did not. Although proven to be highly unreliable [29,30], dogs were breed labelled by an experienced shelter staff member, based on morphological breed characteristics described by the Fédération Cynologique Internationale (FCI), with the aim to match a control group of pet dogs based on body conformation and body weight.

Of the 55 shelter dogs, 38 dogs were adopted (mean = 72 days after intake, range 15–455 [min.-max.] days), 7 dogs returned to their owner, 5 moved to another shelter and 5 dogs were euthanised due to behavioural reasons as decided by a euthanasia commission.

**Matched control group of pet dogs.**   A control group of 21 pet dogs living in their own homes was recruited, percentage-based balanced for characteristics of the shelter dog group based on breed group [31], size (body weight class), age class [13] and sex. Control pet dogs were recruited via social media advertisements and via dog professionals. Owners participated voluntarily, signed an informed consent, and followed their normal routine with their dog during the measurement period.

The pet dog group had a mean age of 3,7 years (range 1–11 years), 8 were female (1 intact, 7 neutered) and 13 were male (2 intact, 11 neutered). Pet dogs were assigned to weight classes: <10kg, n = 6; 10-20kg, n = 6; 20-30kg, n = 8; >30kg, n = 1.

### Housing

Dogs were individually housed in kennels with an in- and outside enclosure (both ~5 m²), separated by a hatch with a plastic flap. The inside enclosure was glass-fronted with a tiled floor and included toys and food and water bowls. The resting place of the dogs, which was a basket with blankets or only a blanket when dogs were not used to a basket, was mostly placed in the inside kennel, and the position of the resting place per dog did not differ over days. The outside enclosure was bar-fronted with a concrete floor and only had a food and water bowl in it.

As a result of the glass-fronted isolated inside enclosure, the inside was less noisy and better temperature regulated than the outside enclosure. Dogs had access to both enclosures for most of the day, except during cleaning (morning). The kennels were accessible by staff and volunteers between 8:00 and 17:00 to care for the dogs. Kennels were not open to the public. Kennels were cleaned once every day between 8:00–13:00. Most dogs were fed dry kibble 2 times a day, some 3 times when needed, and water was available *ad libitum*. In the afternoon, food enrichment was provided for the dogs, (e.g. bones, food puzzles). Dogs were allowed out on a playing field once or twice a day, preferably with other dogs. Fully vaccinated dogs were allowed to walk with volunteers in the area around the shelter for 20–45 minutes every day or every other day.

## Measurement procedures

Dogs were observed during the first two-week acclimatisation period in the shelter for the following variables: nocturnal activity, nocturnal behaviour, urinary cortisol/creatinine ratio, body weight, body condition score and food intake. The timing of sampling is visualized in Fig 1. Sampling variables are described in detail below. The day the dogs entered the shelter was designated as day 0, with night 1 starting at 00:00 after intake, followed by day 1, etc. Hair samples were also collected after intake in the shelter for another study [32].

**Nocturnal activity.** Nocturnal activity was measured using a small light-weight tri-axial accelerometer, the ActiGraph® GT9X Link (3.5 × 3.5 × 1 cm, 14 grams, ActiGraph Corp, USA). Protective hard-plastic cases fitted special for the ActiGraph® were 3D printed at our technical support lab (Service en Onderhoud Bèta, Utrecht University). After intake in the shelter, the dogs were fitted with a collar and an ActiGraph® in its protective case was attached to the dog's collar [33] with duct-tape. Dogs wore the accelerometer 24/7 for the first 14 days, therefore data was collected during the first 13 nights in the shelter. Control pet dogs wore the ActiGraph® for 4 consecutive days.

Nocturnal activity data was read out for 4 hours for every night. For shelter dogs, this period was from 0:00–4:00 AM as this was a period during which no natural light was visible during all seasons and dogs were not disturbed by humans in the shelter. For control pet dogs, a same timeframe of 0:00–4:00 AM was chosen if the owners went to sleep before 11:00 PM. If they went to bed later, one hour after the owners went to bed (logged by the owners on a form) the 4-hour timeframe started.

ActiGraph® data was processed and analysed with the accompanying software, ActiLife® (version 6.13.4; ActiGraph Corp., LLC), using a defined time period of 15 sec (called epochs). In ActiLife®, the following activity measures were calculated for the 4-hour time frame: 1) Vector Magnitude Counts per minute (*VMCpm*); 2) percentage of time spent active (*% active*); 3) number of inactive bouts (*# inactive*); and 4) number of inactive bouts that took longer than 15 minutes (*# inactive >15min*), for descriptions of these measures see Table 1. An active epoch was defined when there were >0 counts, an inactive epoch when there were 0 counts.

**Nocturnal behaviour of shelter dogs.** Nocturnal behaviour of shelter dogs was observed in addition to ActiGraph® measures, to validate accelerometer recordings and observe behavioural indicators of stress during the night. We were not able to observe nocturnal behaviour of control pet dogs due to privacy and practical reasons.

Shelter dogs were monitored using a video recording system (PRO 8-channel camera system, BASCOM, Nieuwegein, the Netherlands) with two infrared bullet night vision cameras per dog, focussed permanently on the outside and inside enclosure of the kennel. Sound recordings per kennel were not available. Video recordings of the dogs from 0:00–4:00 were saved from night 1, 2, 3, 5, 7, 9 and 12 in the shelter (see Fig 1), which matched the

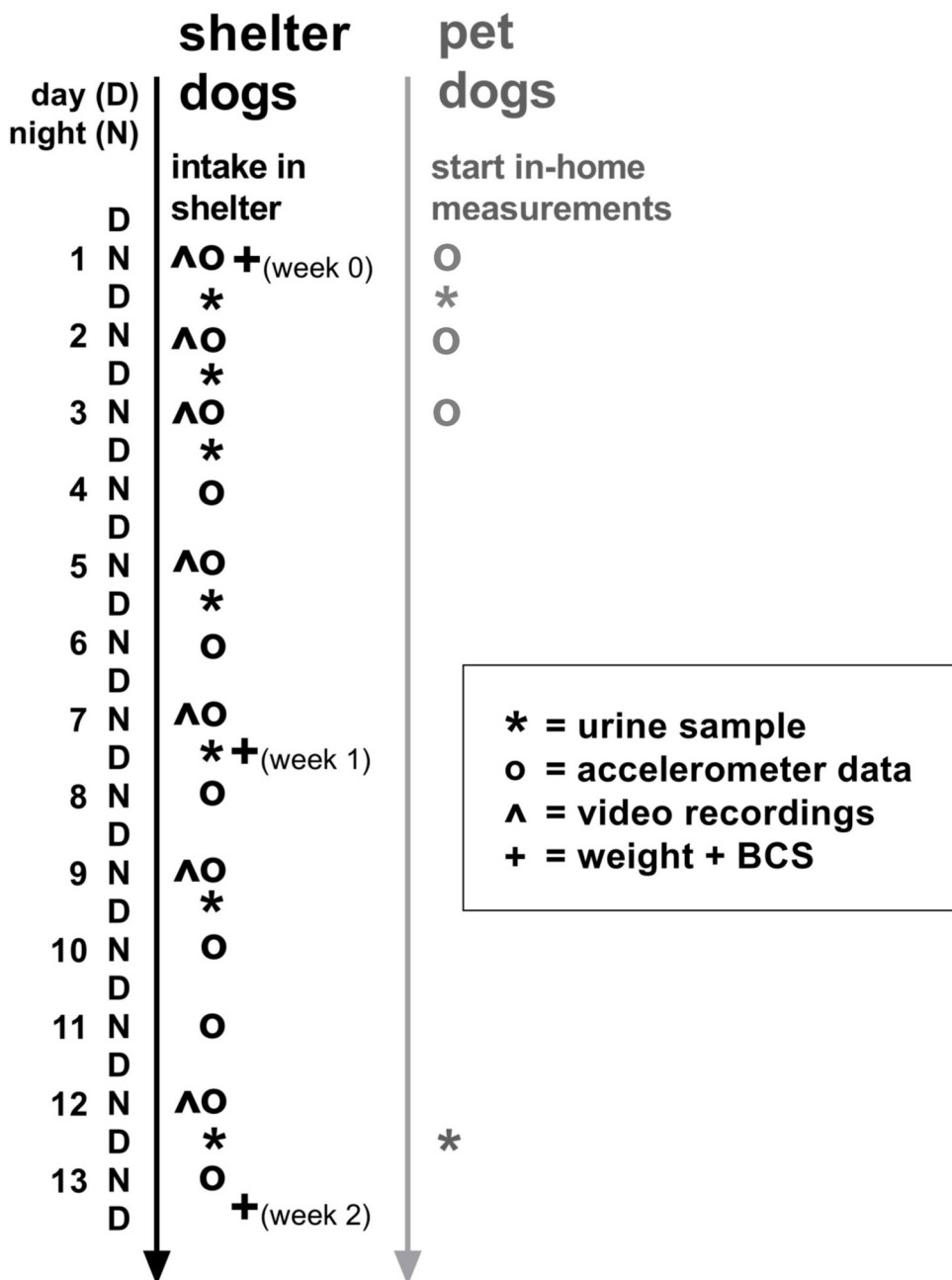

**Fig 1. Timeline with measurement moments for the collected variables in shelter dogs paralleled with the control group of pet dogs.** N = night, D = day. Symbols in the figure represent different types of data sampling: * = urine sample, o = accelerometer data, ^ = video observations, + = weight.

accelerometry recording times. Due to shortage of cameras and technical issues, footage of 6 or all 7 nights were available for 37 dogs (of the 55 dogs) for observations, other dogs were removed from behavioural observations to achieve a relatively complete dataset for the dogs used for analysis.

Nocturnal behaviour was observed *post-hoc* from the video recordings, by two experienced observers using The Observer XT 15 (Noldus Information Technology). Before behavioural

**Table 1. ActiGraph® activity measures as processed and analysed by ActiLife®.**

| ActiLife® measures | Abbreviation | Description | Indicative of: |
|---|---|---|---|
| Vector Magnitude Counts per minute | *VMCpm* | The overall counts divided by the total duration of analysis in minutes | Total activity, both frequency and intensity |
| Percentage of time spent active | *% active* | Summed all 15 second epochs during which activity was determined | Total duration of activity |
| Number of inactive bouts | *# inactive* | Number of bouts in which no activity was determined | Sleep fragmentation and restlessness |
| Number of inactive bouts that took longer than 15 minutes | *# inactive >15 min* | Number of bouts in which no activity was determined, for longer than 15 minutes | How often dogs could fulfil a sleep cycle (of 16–20 minutes [24]) |

observations, inside and outside video records per night per dog were coupled and were randomised in Excel (total = 255 videos) to allow blind scoring for the night in the shelter. The ethogram of the observed behaviour was composed of activity and resting behaviour, and the position in the kennel. Also, several spontaneous behaviours were added to the ethogram, adopted from the literature on behavioural indicators of stress in dogs and kennelled dogs (see Table 2 for references). These behavioural indicators of stress were previously related to acute and/or chronic stress, and are therefore potentially indicative of stress, but also are seen in other (not stress related) contexts. Therefore, we were mainly interested in the changes in these spontaneous behaviours over time in the shelter, simultaneously with the other stress related parameters (UCCR and nocturnal activity). For convenience, we call these behaviours, which are potentially indicative of stress: 'behavioural indicators of stress'.

Inter-observer Cohen's Kappa reliability of the two observers was 0.99 (duration/sequence) and 0.65 (frequency/sequence). After observation of 28 videos, both observers re-observed one of the first videos of these 28 again, to calculate intra-observer Cohen's Kappa reliability. These Cohen's Kappa's were ranging from 0.92–0.99 (duration/sequence) and 0.76–0.90 (frequency/sequence).

**Urinary cortisol/creatinine ratio.** Urinary cortisol/creatinine ratios (UCCR, cortisol corrected by creatinine for dilution effects) were evaluated as a non-invasive measure of stress-induced cortisol responses [39] and thus HPA-axis reactivity in the shelter environment.

Urine samples of shelter dogs were collected on day 1 (day after intake), 2, 3, 5, 7, 9 and 12 after entering the shelter, as we expected most changes to occur during the first days in shelter just as found by Hiby et al. [12] and Van der Laan et al. [13]. To collect the sample, dogs were taken out of their kennels between 7:30–11:30 (median = 8:37) on measurement days by one of the researchers. Naturally voided morning urine was captured mid-stream with a ladle and transferred immediately with a disposable pipette to a polypropylene tube (5 ml 75x13 mm, Sarstedt AG & Co). If the dogs were not naturally urinating outside of their kennel, urine in the in- or outside kennel was sucked up with a pipette and transferred to a tube. In total, 59/349 (= 17%) analysed in-shelter urine samples were collected from the kennel floor. Samples were frozen in a -20°C freezer [8,12] in the shelter within 56 minutes (median = 10 minutes) and transferred to -80°C within 20 days (median = 5 days) after sampling, until analysis.

Owners of control pet dogs were trained to collect urine from their dog by an instruction form and explanatory video. Urine of control pet dogs was collected between 6:20–11:00 (median = 8:30) at day 1 and around day 12 after the start of the measurement period (second collection moment was after a mean of 11 days and a range of 10–14 days, but in this article is called day 12 for convenience and for comparison with day 12 for shelter dogs), by the same method as described above. Owners saved the samples in their own freezer (-10° to -20°C) until a researcher collected the samples within 17 days after sampling and transferred them to -80°C, until analysis.

**Table 2. Ethogram of nocturnal activity and resting behaviour, place in the kennel and behavioural indicators of stress.** Adopted from mentioned references.

| Category | Behaviour | Description | References | Analysis |
|---|---|---|---|---|
| Activity behaviour (*mutually exclusive*) | Recumbent head down | The dog's abdomen is touching the ground with its dorsal, caudal or lateral side whilst legs are extended forwards, curled close to the body or laid to one side. Eyes may be open, closed or not visible. Head is resting on the ground, kennel inventory or paws. | [22,34] | Percentage of time Rate per minute of transitions in the different active behaviours (recumbent head up/ recumbent head down/stationary/ movement) |
| | Recumbent head up | As above, but with its head up and held above the ground. Eyes can be open or closed. | | |
| | Stationary | Sit: hindquarters in contact with the ground and front legs extended being used for support; or stand: four feet in contact with the ground and legs fully, or almost fully, extended supporting the body. | [9,35] | |
| | Movement | Dog moves around the enclosure (e.g. walking, running, mobile exploration). Ambulates at any speed. | [9,36] | |
| Position in kennel (*in- and outside kennel are mutually exclusive*) | Resting place | Dog resides in dog bed, dog crate (bench) or on blanket (if no bed or crate is available). | | Percentage of time |
| | Inside kennel | Dog('s head) resides in the inside kennel. | | |
| | Outside kennel | Dog('s head) resides in the outside kennel. | | |
| Oral behaviour | Lip/snout licking | Tongue protrudes and licks own lips or snout. | [9] | Rate per minute when in sight of the camera and dog was active (head up, stationary, movement) |
| | Yawning | (Slowly) opens mouth wide and closes eyes. | | |
| | Panting | Breathes deeply and quickly with mouth open and tongue (often) hanging out. | | Proportion of time when in sight of the camera and dog was active (head up, stationary, movement) |
| | Drinking | Laps water from water provision/bowl. | | |
| Body behaviour | Autogrooming | Behaviours directed towards the subject's own body, like scratching, licking and biting-self. | [7] | |
| | Paw lifting | Raises single forepaw while sitting or standing and holds it above the ground. Except during barking or whining when a dog briefly lifts one of its paws while sitting or standing due to lifting the head upwards to vocalise. | [9] | Rate per minute when in sight of the camera and dog was active (head up, stationary, movement) |
| | Body shake | Shakes whole body, including head, rapidly from side-to-side. Not: trembling. | | |
| | Repetitive locomotor behaviour | Repetitive behaviour: motions that were repeated ($\geq$ 3 times) with minimal interruptions. Circling: Repetitive circling around pen; Tail chasing: Repetitive chasing of tail; Pacing: Repetitive pacing usually along a fence; Wall bounce: Repetitive jumping at wall, rebounding off it. | [37,38] | |

Analysis took place at the veterinary diagnostic laboratory of the Faculty of Veterinary Medicine at the Utrecht University, the Netherlands. Cortisol was analysed with a Radio-Immuno-Assay [40] and creatinine was analysed using addition of picric acid and spectrophotometry with Jaffé calculation. The UCCR was calculated and expressed as: cortisol (nmol/l): creatinine (μmol/l) x1000 = ratio x10^-6.

**Weight & body condition score (BCS).** As body weight loss can be stress-related [41] and because we observed a loss in body weight in shelter dogs before [13], dogs were weighed on a scale in the shelter (AllScales® Europe) by the veterinarian or one of the researchers on days 1, 7 and 12 (week 0, 1 and 2).

Body weight loss can be a consequence when overweight dogs enter the shelter and receive less food and/or a better quality of food than usual with their previous owners. As obesity is associated with higher cortisol levels in both humans and animals [42], we additionally evaluated the BCS of the dogs at intake and after 2 weeks. BCS was scored by the trained researchers or veterinarian on a 9-point scale from 1 (emaciated) to 9 (obese), with 4 and 5 representing

ideal body condition, as developed and validated by Laflamme [43]. Interobserver agreement of three researchers was set at 80% during a pilot, with deviations of maximum 1 BCS point.

**Food intake.**   To evaluate food intake of the dogs in the shelter, the amount of food eaten was noted [9] during the first two weeks after every meal. Every morning and afternoon, right before the next feeding moment, one of the researchers noted whether the food bowl of the dog was empty (almost or all food was eaten), not empty but not full (where some food was eaten but also some was left) or still full (almost none eaten).

## Statistical analysis

Data were stored and cleaned in Microsoft Excel® (Microsoft Corporation). Statistical software program RStudio (version 1.3.1093 –©RStudio, Inc.) was used to perform linear mixed model analysis with the package 'Nlme' [44], exploratory graphs with packages 'ggplot2' and 'ggpubr'. Graphs in this paper were created in Graphpad Prism (version 9 –©GraphPad Software, LLC).

Data were explored for methodological outliers by a 'mean ± 3 standard deviations' cut-off. One UCCR datapoint was removed (one shelter dog, 6 weeks in shelter datapoint), 9 outliers were not removed as these belonged to dogs that had high UCCR's in general and therefore stood not out on individual dog-level. For nocturnal activity and behaviour no outliers were removed, as all outliers were confirmed not to be methodological errors during the evaluation of behaviour and activity observations, and outliers of behaviour and nocturnal activity were re-evaluated in the observations and measurement data.

Relative changes in body weight were calculated for within-subjects analysis as proportional body weight for week 1 (day 7) and week 2 (day 12) based on the weight in week 0 (day 1) = 1.0 = 100%. Food intake was scored as 0 (full bowl, almost no food was eaten), 1 (not empty but not full bowl, some food was eaten but some was left) or 2 (empty bowl, almost all or all food was eaten) per meal. The mean food intake of the dog was calculated over the two weeks of observation (26 observation moments). Dogs were labelled as a 'low eater' (mean <1) or a 'medium-good eater' (mean 1–2). Behavioural indicators of stress shown in <10% of the dogs (= 4 dogs) on all days were excluded from analysis, as these indicators provided insufficient data for analysis [9,12].

Outcome variables UCCR, activity measures, behaviour variables and proportional body weight were evaluated for normality with Shapiro-Wilk tests and visual evaluation of boxplots and quantile-quantile plots of the data. UCCRs, all activity variables except # *inactive >15 min*, and all behavioural variables were skewed and were therefore (natural) log transformed before statistical testing and back transformed for interpretation. Back transformed (exp) log model values resulted in ratios, with a ratio <1 meaning a lower value and >1 a higher value than the reference mean. Alpha level was set at p<0.05. For mixed models, 95% confidence intervals (CI) ranges <1 or >1 were considered significant.

Linear mixed models were fit for each outcome variable (see below for which variables). Fixed effects were added to each model: 'day' (UCCR)/'night' (activity measures)/'week' (body weight/BCS), 'kennel history' (no/unknown/yes), 'body weight class' (<10 kg, 10–20 kg, >20–30 kg, >30kg), 'body condition score at intake at the shelter' (underweight BCS 1-3/ideal weight BCS 4-5/overweight BCS 6-9), 'age class' (1–4 years, 5–7 years, 8–13 years), 'sex' (male/female), 'neuter status' (no/unknown/yes), 'reason for admission to the shelter' (relinquished/stray/crisis pension), 'total time spent in shelter' (short <6 wks/medium 6–12 wks/long >12 wks/euthanasia in shelter/recollected by owner or moved to other shelter), and 'food intake' (low eater/medium-good eater). 'Day', 'night' or 'week' was included as a factor and not treated as continuous in the model. Interactions between 'day', 'night' or 'week' and one of the

other main factors were added in the start model when visual inspection of boxplot graphs revealed potential interactions. Full models were tested with a random effect for 'dog ID' (individual identity) and explanatory variables were dropped based on a backward selection approach, using the Akaike information criterion (AIC) to determine the best model fit with maximum likelihood estimation. 'Day', 'night' or 'week' were never dropped, as this was the main factor of interest. Explanatory variables (factors) included for best fit are described in the results section per model. Various correlational and variance structures (with autoregressive model of the order 1 (AR1) correlation structure or weights) were added on the final model to test for the best fit. With the best fitting model structure, restricted maximum likelihood estimation was used for the final model. Models were evaluated by visual inspection of the residuals (normality and constant variance).

Pet dogs and comparisons with shelter dogs: UCCR and nocturnal activity levels of the shelter dogs were compared to pet dog data (between-subjects) and pet dog difference between days (within-subjects) using two-tailed t-tests on log transformed data using Bonferroni corrections for multiple comparisons. For UCCR levels, the following comparisons were made: pet dogs day 1 versus day 12, pet dogs day 1 versus shelter dogs day 1, and pet dogs day 12 versus shelter dogs day 12. For nocturnal activity levels, the following comparisons were made: pet dogs night 1 versus night 3, and pet dogs night 3 versus shelter dogs night 12. To compare differences in nocturnal activity in shelter versus pet dogs during the first 3 nights of measurement (night 1–3), models were also fit for each activity outcome with a fixed effect for 'night' and 'group' (shelter dogs/pet dogs) and an interaction between the two, and a random effect for 'dog ID'. No extra factors were added as both groups were percentual matched on most factors. For *VMCpm* and *% active*, models with a random intercept and slope fitted best, for *# inactive* a model with only random slope fitted best and for *# inactive >15 min* a model with only random intercept fitted best. Results are reported with sample estimated mean difference (semd) or sample estimated mean difference in ratio (semd[r]) and with 95% confidence interval (95% CI).

Shelter dogs—changes over days in shelter: Linear mixed effects models were fit for each outcome variable for within-subjects analysis.

For *UCCR levels*, a model with a random intercept, 'weights = varIdent' for 'day', and correlational structure CAR1 fitted best.

For nocturnal activity levels *VMCpm*, *% active*, *# inactive*, and *# inactive >15 min*, a model with random intercept and 'weights = varPower' for 'night' fitted best. For *% active*, a correlational structure CAR1 was also added for best fit. For *body weight proportional change*, a model with only a random slope and no other structures fitted best. To evaluate changes in BCS over time in the shelter, BCS at intake and after 13 days were compared for dogs that had both values using a two-tailed paired t-test, as data were normally distributed. For nocturnal activity behaviour in the shelter, linear mixed models were also fitted on the variables *% recumbent head up*, *% stationary*, *% of movement*, *rate per minute of transitions in the different activity behaviours (recumbent head down/recumbent head up/stationary/movement)*, and *% of active behaviour*, where all active behaviours were grouped. For all activity behaviour variables, a model with a random intercept and correlational structure CAR1 fitted best. As dogs with an unknown neuter status were only 3 bitches, results are not presented here. These dogs were deviant on several activity measures from dogs that were neutered or intact, but conclusions are difficult to draw as these bitches could both be neutered or intact, and differences based on sex was included in the 'sex' variable.

Concerning behavioural indicators of stress, if the dogs would rest or sleep more during the night, logically they would show less behavioural indicators of stress, as the dogs simply don't show these behaviours during rest or sleep. Therefore, we calculated the proportion of time

and rate per minute (RPM) of these behaviours when dogs were in-sight of the camera [9] and the dog was active (during activity behaviours: recumbent head up, stationary or movement). *Repetitive locomotor behaviour* was only seen once in one dog, for 14 seconds during the second night in the shelter, and was therefore excluded from analysis. Some nights, *panting* and *drinking* were seen in less than 4 dogs (<10%) and therefore these behaviours were excluded from analysis. *Paw lifting* was seen irregularly: in >4 dogs at all nights but 11 dogs did not show *paw lifting* at any night and were therefore excluded. *Paw lifting* was therefore excluded from behaviour analysis, but results are shown in the results section with sample size per night. *Lip/snout licking* and *yawning* were combined into one 'oral' category (RPM) to gain enough data per dog per night. As a result, all dogs showed *oral behaviour*, *autogrooming*, and *body shaking* for at least one observed night and sample sizes per night were high enough to perform mixed model analysis. However, *autogrooming* had too many zero values to perform analysis as dogs did not show the behaviour at these days, therefore all values were added +1 before log-transforming (log[x+1]) to normality. Linear mixed models with only random intercept were fitted best on the variables: *oral behaviour* with 'weights = varIdent' for 'night', *autogrooming* with 'weights = varPower' for night, and *body shake* with 'weights = varPower' for 'night'.

## Ethics

The study protocols were submitted to the institutional committee Utrecht Animal Welfare Body of Utrecht University, The Netherlands. The Animal Welfare Body concluded that the study does not meet the definition of an animal experiment as defined in the Dutch Experiments on Animals Act and Directive 2010/63/EU, as the animals would encounter minimal levels of discomfort and the observations were conducted within the normal routine of the animal shelter. The participating animal shelter consented to the study. Owners of the pet dogs agreed and volunteered to participate in this study and signed informed consent for participation and publication of the results, conforming to the General Data Protection Regulation in the Netherlands.

## Results

### Nocturnal activity (accelerometer output)

**Shelter dogs compared to pet dogs.** For the pet dogs, no significant differences were found between night 1 and 3 for all nocturnal activity accelerometer outputs (two-tailed t-tests on log transformed data, with *VMCpm*: semd[r] = 0.86, 95% CI = 0.62–1.18, t[39] = -0.97, p = 0.34; *% active*: semd[r] = 0.93, 95% CI = 0.73–1.18, t[39] = -0.64, p = 0.53; *# inactive*: semd[r] = 0.94, 95% CI = 0.79–1.12, t[39] = -0.75, p = 0.46; *# inactive >15 min*: semd = 0.31, 95% CI = -0.60–1.23, t[39] = 0.70, p = 0.49; see Table 3 for original means and standard deviations).

When comparing nocturnal activity of pet and shelter dogs during the first three nights, an interaction between night and group (pet versus shelter) significantly explained all nocturnal activity accelerometer outputs variabilities (mixed model, Table 3). Shelter dogs had higher *VMCpm*, were higher *% active* and had a higher *# inactive* (meaning shorter inactive bouts in general) and lower *# inactive >15 min* than pet dogs on all the three nights, with differences between the groups being larger on night 1 than on night 3 (Fig 2). When comparing shelter dogs night 12 with pet dogs night 3, shelter dogs still had significantly higher *% active*, but not *VMCpm*, *# inactive* and *# inactive > 15 min* (two-tailed t-tests on log transformed data, *VMCpm*: semd[r] = 0.71, 95% CI = 0.49.-1.01, t[63] = -1.91, p = 0.06; *% active*: semd[r] = 0.66, 95% CI = 0.51–0.86, t[63] = -3.13, p = 0.0026; *# inactive*: semd[r] = 0.91, 95% CI = 0.76–1.09, t[63] = -0.99, p = 0.32; *# inactive >15 min*: semd = 0.38, 95% CI = -0.49–1.25, t[63] = 0.87,

**Table 3. Original means (±SD), and model results for nocturnal activity accelerometer outputs for the shelter dog group compared to the pet dog group, during the 4 hour measurement period.**

| Accelerometer output | Night | Original values | | Model results | | | | | |
| | | Original means ± SD | | Estimated | | Conditional F-test | | | |
| | | Shelter dogs | Pet dogs | EP | 95% CI | F | NumDF | DenDF | Sign. |
|---|---|---|---|---|---|---|---|---|---|
| *VMCpm* | 1 | 350 ± 363 | 42 ± 15 | 0.17 | 0.11–0.28 | 18.765 | 3 | 124 | < .0001 |
| | 2 | 232 ± 236 | 62 ± 52 | 0.35 | 0.22–0.54 | | | | |
| | 3 | 141 ± 124 | 54 ± 43 | 0.45 | 0.30–0.68 | | | | |
| | 12 | 86 ± 81 | | | | | | | |
| *% active* | 1 | 28% ± 21% | 6% ± 2% | 0.24 | 0.17–0.35 | 20.923 | 3 | 124 | < .0001 |
| | 2 | 20% ± 12% | 6% ± 3% | 0.38 | 0.27–0.52 | | | | |
| | 3 | 15% ± 12% | 6% ± 3% | 0.46 | 0.34–0.63 | | | | |
| | 12 | 10% ± 6% | | | | | | | |
| *# inactive* | 1 | 49 ± 21 | 28 ± 8 | 0.56 | 0.45–0.70 | 9.476 | 3 | 124 | < .0001 |
| | 2 | 43 ± 19 | 29 ± 6 | 0.72 | 0.60–0.86 | | | | |
| | 3 | 38 ± 14 | 29 ± 7 | 0.80 | 0.67–0.95 | | | | |
| | 12 | 33 ± 13 | | | | | | | |
| *# inactive >15 min* | 1 | 3.4 ± 2.0 | 6.7 ± 1.6 | 3.35[1] | 2.37–4.33[1] | 18.097 | 3 | 124 | <0.0001 |
| | 2 | 4.4 ± 1.9 | 6.6 ± 2.0 | 2.21[1] | 1.25–3.17[1] | | | | |
| | 3 | 5.1 ± 1.9 | 6.4 ± 1.3 | 1.26[1] | 0.29–2.23[1] | | | | |
| | 12 | 6.0 ± 1.7 | | | | | | | |

Estimated (EP) mean of shelter dogs compared to mean in pet dogs at same day, and 95% confidence intervals (CI). Reference category in the model was the shelter dog group, therefore all estimates are ratios of estimated mean of pet dogs compared to the estimated mean of shelter dogs at the same night (except for *# inactive >15 min*, see footnote). Conditional F-testing revealed F, DF's and significance of interaction night * group in the model.

[1] Estimated parameter and 95% confidence intervals represent the difference between the pet dog value compared to the shelter dog estimated parameter in actual values, as *# inactive >15 min* was not log transformed.

p = 0.39; see Fig 2). Also, interindividual variability was higher in shelter dogs than pet dogs (Fig 2A–2C) and was highest during the first nights in the shelter.

**Shelter dogs—changes over days in shelter.** *VMCpm.* The main effects night and neuter status significantly explained *VMCpm* variability (see Fig 2A and S2 Table). *VMCpm* at night 3–13 were significantly lower than at night 1.

*% active*: The main effects night and neuter status significantly explained *% active* variability (see Fig 2B and S3 Table). *% active* at night 2–13 was significantly lower than at night 1.

*# inactive*: The main effects night, kennel history and weight class significantly explained variability for *# inactive* (see Figs 2C and 3 and S4 Table). Considering night, *# inactive* at night 1 was significantly higher than at night 2–13. Dogs that had a known history in a kennel environment had a lower *# inactive* than dogs that did not have a history in a kennel environment and dogs for which kennel experience was not known. For weight class, weight class 10–20 kg (n = 13) had significantly higher *# inactive* than weight classes >20–30 kg (n = 12) and >30 kg (n = 11). Weight class <10 kg (n = 19) also had a significantly higher *# inactive* than weight class >30 kg.

*# inactive >15 min*: The main effects night and weight class significantly explained variability in the *# inactive >15 min* (see Figs 2D and 3 and S5 Table). *# inactive >15 min* increased over the nights, with night 1 being significantly lower than nights 2–13. Weight class >30 kg had significantly higher *# inactive >15 min* than weight class <10 kg.

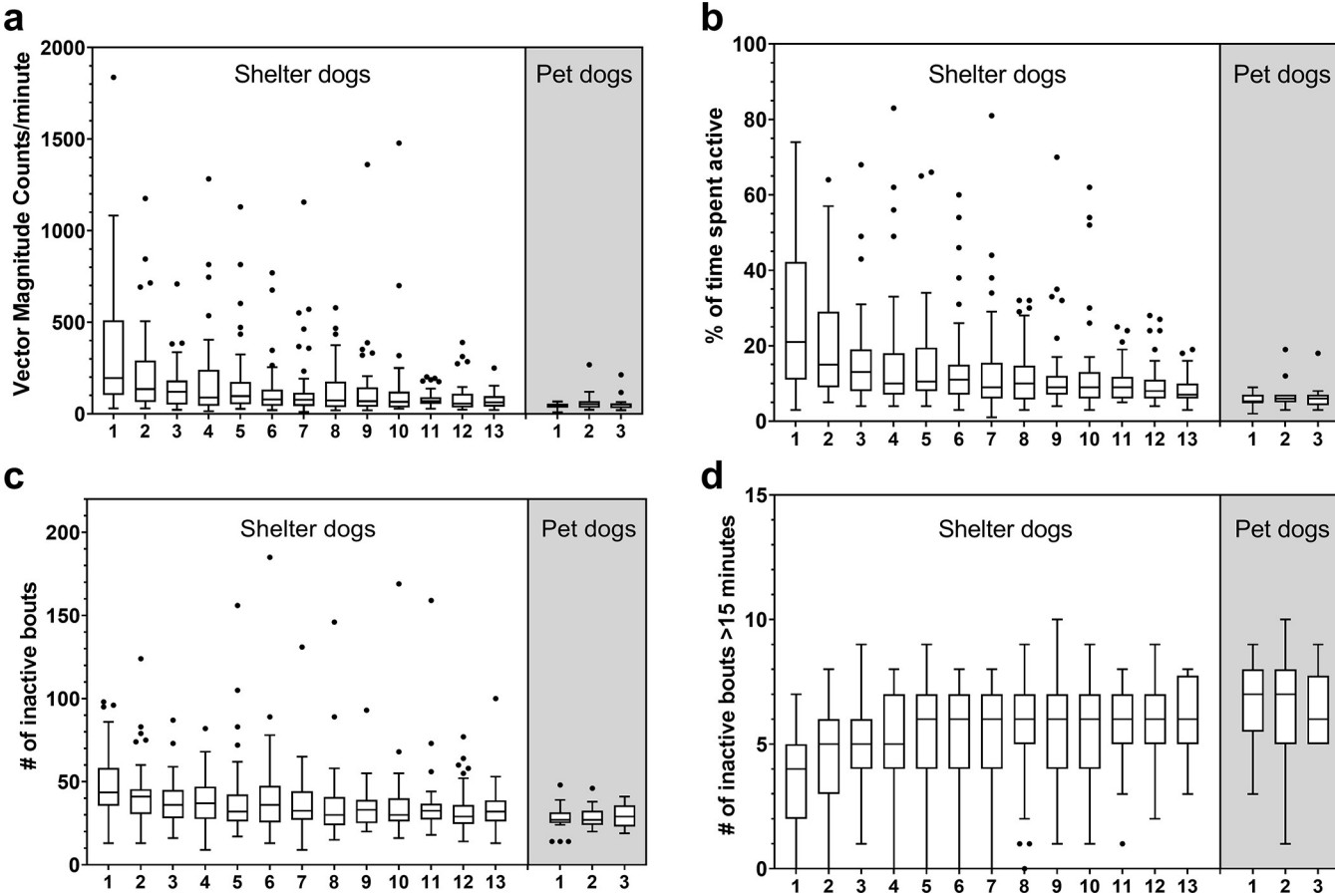

**Fig 2. Nocturnal activity accelerometer results.** Box and whisker (Tukey) plots with median and outliers (dots) for the shelter dog group (white area) on day 1, 2, 3, 5, 7, 9 and 12 in the shelter (x-axis) and for the control pet dog group (grey area) on day 1, 2 and 3 (x-axis) into the study, during the 4 hour measurement period. a) *VMCpm*, b) *% active*, c) *# inactive*, d) *# inactive >15 min*.

## UCCR

**Shelter dogs compared to pet dogs.**   No significant difference was found between UCCR of control pet dogs on day 1 and day 12 (t-test on log transformed data, semd[r] = 1.02, 95% CI = 0.72–1.36, t[40] = -0.10, p = 0.92).

UCCRs of the shelter dogs were significantly higher than those of the pet dogs both on day 1 and 12, although the difference on day 12 was smaller (t-tests on log transformed data, day 1: semd[r] = 2.76, 95% CI = 1.96–3.87, t[58] = 5.97, p < 0.001; day ~12: semd[r] = 2.05, 95% CI = 1.47–2.87, t[63] = 4.29, p < 0.001, Fig 4). Also, interindividual variability in UCCR responses was higher in sheltered dogs than in pet dogs.

## Shelter dogs—changes over days in shelter

The main effects day, weight class and kennel history all significantly explained UCCR variability (mixed models, see Figs 3 and 4 and S6 Table for model results). Considering day, UCCRs at day 9 and 12 were significantly lower than at day 1. Dogs in lower weight classes (<10 kg and 10–20 kg) had significantly higher UCCR than dogs in the two higher weight classes (>20–30 kg and >30 kg). Dogs that had stayed in a kennel environment before had significantly lower UCCRs than dogs that had not stayed in a kennel before, with intermediate

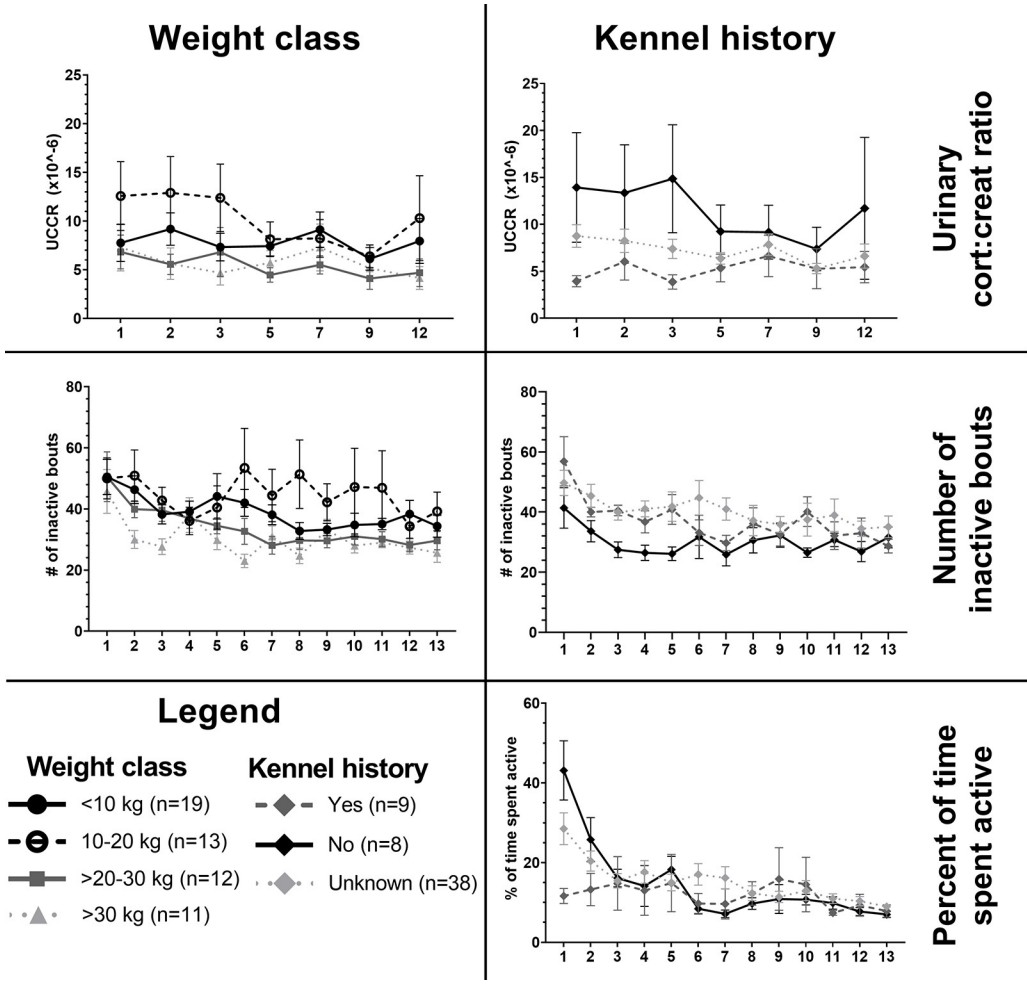

**Fig 3. Nocturnal activity of shelter dogs divided for main effects Weight class and Kennel history for UCCR and nocturnal activity accelerometer outputs # inactive and % active during the night (0:00–4:00).** Means and standard error of the mean (SEM) for the shelter dog group on night 1, 2, 3, 5, 7, 9 and 12 (x-axis) in the shelter, divided in different body weight classes and kennel history classes (yes = had been in a kennel environment before, no = had not been in a kennel environment before, unknown = kennel history was unknown).

UCCRs for the group of which the history of kennel experience was unknown (largest group of dogs).

### Behaviour

**Place in kennel.** Shelter dogs spent on average 71% of the total time (0:00–4:00) on their resting place during night 1, which increased to 91.8% during night 12 in the shelter (Fig 5). Shelter dogs spent on average 94.3% of the total time in the inside kennel during night 1, and 96.3% during night 12.

**Activity behaviour.** For *% (of total time spent on) active behaviour*, where all active behaviours except recumbent head down were grouped, the main effects night, weight class, sex and kennel history significantly explained variability in the percentage of active behaviour (mixed models, see Figs 6 and 7 and S7 Table). Also, interindividual variability of % of active behaviour decreased over time (Fig 6). When these active behaviours were divided in recumbent head up, stationary, and movement (see ethogram in Table 2), the following models were fitted

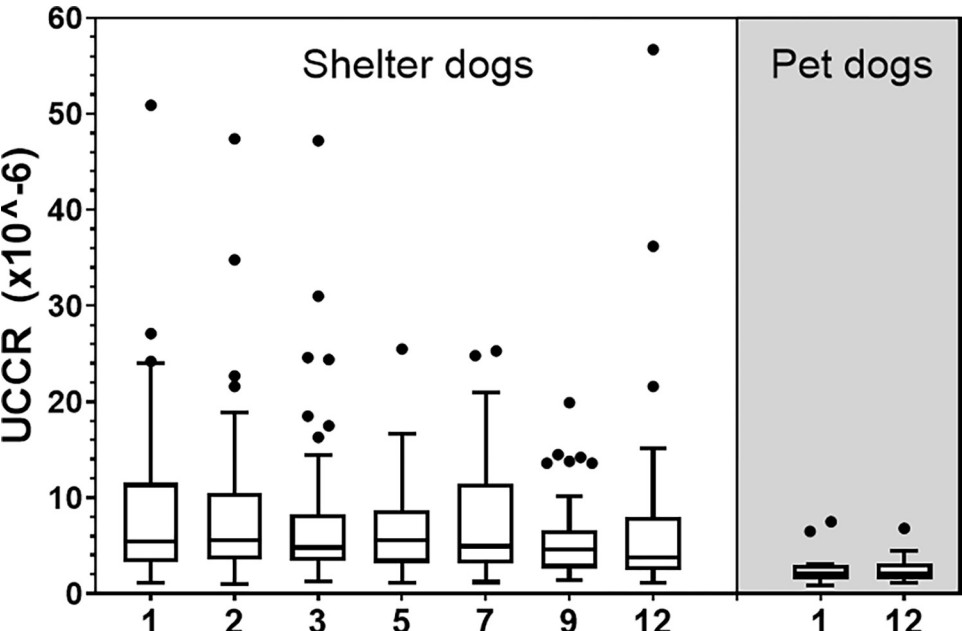

**Fig 4. Urinary cortisol/creatinine ratio (UCCR) results.** Box and whisker (Tukey) plot with median and outliers (dots) for the shelter dog group (white area) on day 1, 2, 3, 5, 7, 9 and 12 (x-axis) in the shelter and for the control pet dog group (grey area) on day 1 and ~12 (x-axis) into the study.

best (S8–S11 Tables). For *% recumbent head up*, the best model included significant factors night, weight class and kennel history. For *% stationary*, the best model included night, weight class, age class and an interaction between night and kennel history. For *% of movement*, the

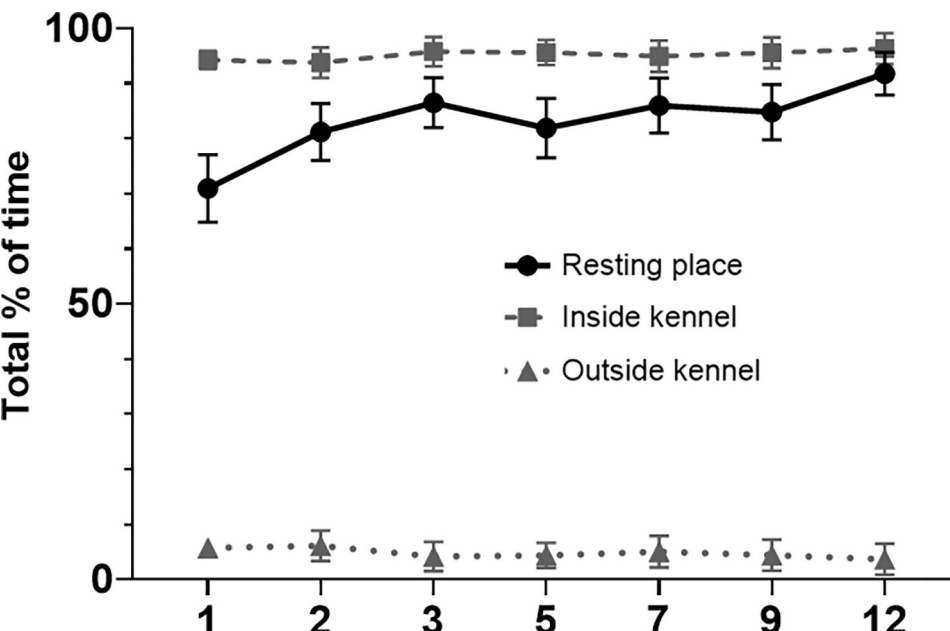

**Fig 5. Total % of time spent on resting place, and inside kennel or outside kennel.** Means and standard error of the mean (SEM) for the shelter dog group on day 1, 2, 3, 5, 7, 9 and 12 (x-axis) in the shelter. Inside and outside kennel were mutually exclusive, but not resting place. Resting place (basket or blanket) was mostly placed in the inside kennel.

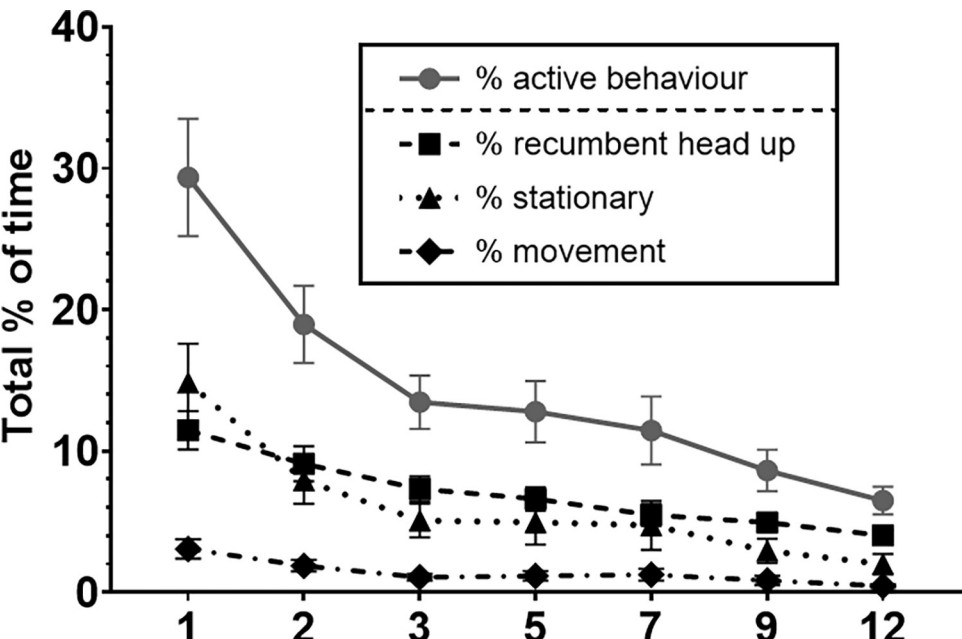

**Fig 6. Total % of time spent showing active behaviours during the night (0:00–4:00).** Means and standard error of the mean (SEM) for the shelter dog group on night 1, 2, 3, 5, 7, 9 and 12 (x-axis) in the shelter, for all active behaviour summed and divided in separate active behaviours (recumbent head up, stationary and movement).

best model included night and neuter status. In addition, best model for the *rate per minute of transitions in the different activity behaviours* included night and age class. The effect of these factors is described in more detail below and visible in Fig 7, including sample sizes per factor variable as the total group of video observed dogs was a subset (n = 37) of the total group of accelerometer and UCCR observed dogs (n = 55).

**Night.**  For all activity behaviour variables, night 1 was higher than later nights. During night 1, the *% of active behaviour* was higher than during night 2, 3, 5, 7, 9 and 12. The *% of recumbent head up* and *% of movement* was higher during night 1 than night 3, 5, 7, 9 and 12, but not night 2. In addition, the *rate per minute of transitions in the different activity behaviours* was higher on night 1 than night 3, 9 and 12.

**Weight class.**  Smaller dogs (<10 kg, n = 10; and 10–20 kg, n = 11) had a higher *% of active behaviour* and *% of stationary behaviour* than larger dogs of >30kg (n = 7). Dogs of 10–20 kg had higher *% of recumbent head up* than all other weight classes, including dogs <10 kg.

**Kennel history.**  Dogs that had stayed in a kennel environment before (n = 5) had significantly lower *% of active behaviour*, including *% recumbent head up* and *% stationary* behaviour (the latter mainly during night 1, 2, and 5), compared to dogs that had not stayed in a kennel before (n = 7) or dogs for which the history of kennel experience was unknown (n = 25).

**Age class.**  Younger dogs (1–4 years, n = 29) had a higher *% stationary* and *rate per minute of transitions in the different activity behaviours* than middle-aged dogs (5–7 years, n = 6).

**Sex.**  Females (n = 15) showed a higher *% of active behaviour* than males (n = 22).

**Neuter status.**  Not displayed as only 3 bitches had unknown neuter status (see statistics section).

**Behavioural indicators of stress.**  For behavioural indicators of stress, the following models were fitted best (see Fig 8 and S12–S14 Tables).

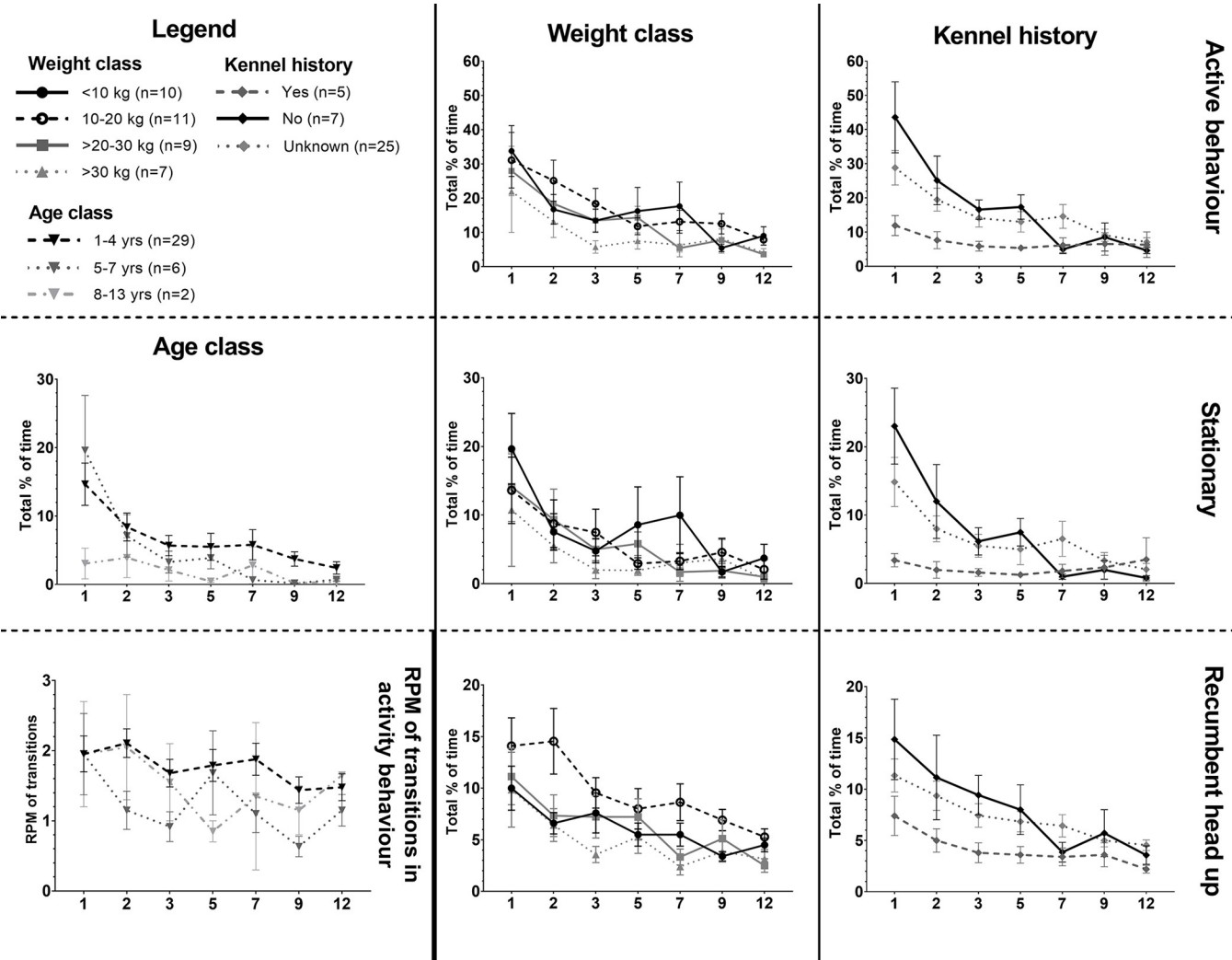

**Fig 7. Nocturnal activity behaviour results.** Total % of time spent showing active behaviours during the night (0:00–4:00), divided for main effects Weight class, Age class and Kennel history, for active behaviour in total, stationary behaviour, recumbent head up behaviour and rate per minute (RPM) of transitions in activity behaviour. Means and standard error of the mean (SEM) for the shelter dog group on day 1, 2, 3, 5, 7, 9 and 12 (x-axis) in the shelter.

**Oral behaviour (yawning + lip/snout licking).** Oral behaviour did not significantly change over nights in the shelter, and no other factors were included in the best fitting model.

**Autogrooming.** The best fitting model included an interaction between night and weight class, and night and age class. Overall, smaller shelter dogs (<10 kg, n = 10, and 10–20 kg, n = 11) showed less autogrooming during the first night in the shelter compared to larger dogs (>20–30 kg, n = 9, and >30 kg, n = 7), and <10 kg also lower than dogs >30 kg on the second night, and autogrooming increased from night 1 and 2 to later nights. For larger dogs no significant difference in autogrooming was found over the days. Older dogs (8–13 years, n = 2) showed more autogrooming on night 1 but less autogrooming on night 7 and 9 compared to younger dogs (1–4 years, n = 29, and 5–7 years, n = 6), but as we only had behaviour data from two older dogs, results are not presented in Fig 8.

**Body shake.** The best fitting model included night, weight class and kennel history. During the first night, shelter dogs significantly showed less body shaking compared to all other nights. Dogs of weight class of >20–30 kg (n = 9) showed more body shaking than dogs of

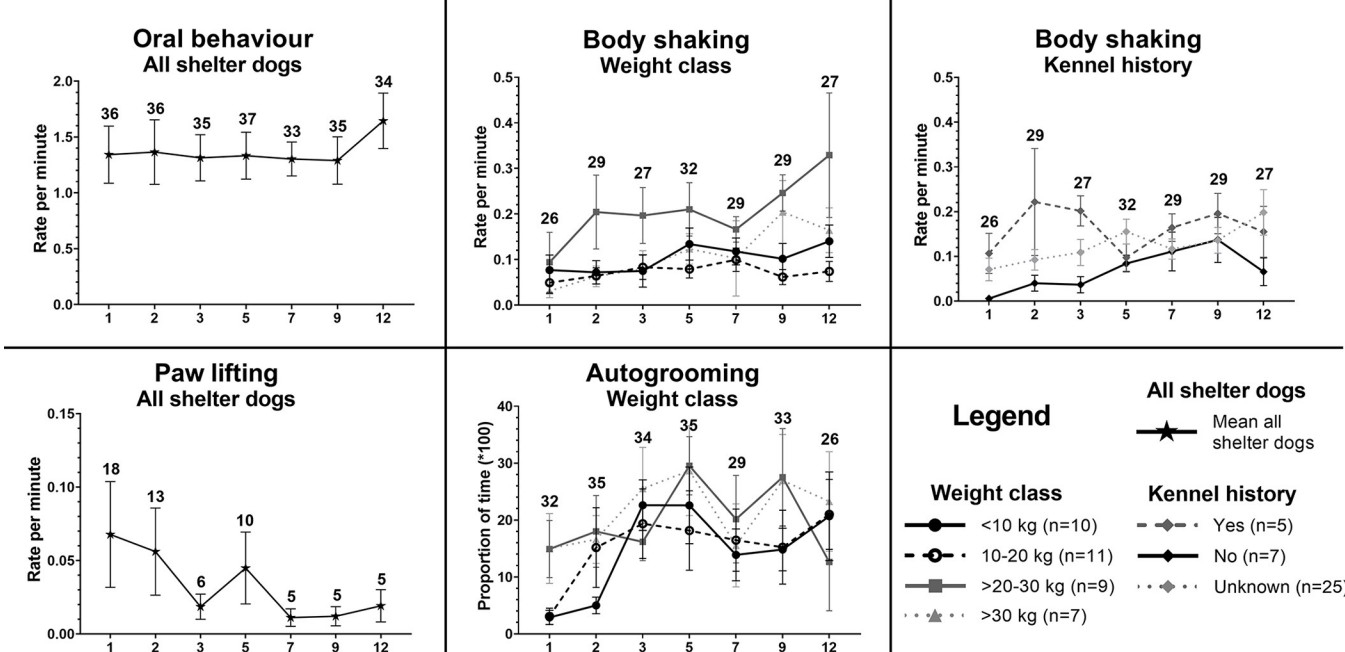

**Fig 8. Nocturnal behavioural indicators of stress results.** Means and standard error of the mean (SEM) for oral behaviour, paw lifting, autogrooming and body shaking, divided for main effects weight class and kennel history where applicable, for the shelter dog group on day 1, 2, 3, 5, 7, 9 and 12 (x-axis) in the shelter. Rate per minute or proportion (x100) of time spent showing these behaviours during the time when shelter dogs were active and in-sight of the camera during the night (0:00–4:00). The total number of dogs that showed the behaviour each night are mentioned above the error bars. Overall sample sizes per weight class and kennel history class are mentioned in the legends.

other body weight classes. Dogs with no kennel history (n = 7) showed less body shaking than dogs with a known history in kennels (n = 5) and dogs with unknown kennel history (n = 25).

**Paw lifting.** Paw lifting (total n = 26) was seen in 18 dogs during the first night and only in 5 dogs during the last nights (night 7, 9 and 12), see Fig 8, resulting in the decreasing mean

**Table 4. Model results for proportional body weight change.**

| Category | | | EPW | 95% CI | Conditional F-test | | | |
|---|---|---|---|---|---|---|---|---|
| | | | | | F | NumDF | DenDF | Sign. |
| Week | Week 1 | | 0.981 | 0.968–0.994 | 18.15 | 1 | 41 | 0.0001 |
| | Week 2 | | 0.957 | 0.941–0.973 | | | | |
| Food intake | Low eater versus medium/good eater | | -0.035[1] | -0.059 --0.011 | 9.36 | 1 | 46 | 0.0037 |
| Age class | 5–7 yrs versus 1–4 yrs | | 0.007[2] | -0.017–0.030 | 2.82 | 2 | 46 | 0.0701 |
| | 8–13 yrs versus 1–4 yrs | | -0.040[2] | -0.076 --0.004 | | | | |
| Week * reason for admission | Stray versus relinquished | Week 1 | 0.010[3] | -0.010–0.030 | 4.43 | 4 | 41 | 0.0046 |
| | | Week 2 | 0.041[3] | 0.016–0.066 | | | | |
| | Crisis pension versus relinquished | Week 1 | 0.014[3] | -0.023–0.051 | | | | |
| | | Week 2 | -0.016[3] | -0.079–0.048 | | | | |

Estimated proportional weight (EPW) and 95% confidence intervals (CI) of proportional body weight change in comparison with week 0 (day 1 after intake, proportional body weight = 1.000 for all dogs), and other factors in the model which significantly explained proportional body weight change. Conditional F-testing revealed F, DFs and significance of the different terms in the model.

[1] Estimated mean difference between the proportional body weight of low eaters compared to reference medium/good eaters on both weeks (1 and 2).

[2] Estimated mean difference between the proportional body weight of the mentioned age class versus reference age class 1–4 years old on both weeks.

[3] Estimated mean difference between specified reason for admission to the shelter and mean in reference reason for admission to the shelter at same week.

and SEM of RPM over nights, as the RPM of dogs that did not show this behaviour was calculated as 'zero'. The 5 dogs during the last nights were different dogs each night.

## Weight, BCS and food intake

Of the total 55 sheltered dogs, 32 dogs were overweight (BCS>5, 58.2% of total) when entering the shelter, 20 dogs had an ideal weight (BCS = 4–5, 36.4%) and 3 dogs were underweight (BCS<4, 5.5%). BCS at intake of relinquished and stray dogs did not significantly differ (mean = 5.4 versus 5.2, respectively, two-tailed t-test: semd = 0.19, 95% CI = -0.35–0.74, t[49] = 0.70, p = 0.48). Thirteen shelter dogs were labelled as 'low eaters' (23.6%), 42 dogs as 'medium/good eaters' (76.4%). The food intake labels did not significantly differ between stray and relinquished dogs (X2 (1, n = 51) = 0.15, p = 0.70). Of the 21 pet dogs, 3 dogs were overweight (14.3%), 17 had an ideal weight (81%) and 1 was underweight (4.8%).

After 12 days in the shelter (n = 46 dogs left), 37 dogs (80%) lost body weight compared to their body weight at intake, 5 dogs (11%) gained weight and 4 dogs (9%) did not change in body weight. For the sheltered dogs, the main effects week, food intake, age class and an interaction between week and reason for admission to the shelter significantly explained proportional body weight variability (see Table 4 and Fig 9A–9D). Proportional body weight decreased significantly from week 0 to week 1 and week 2. Dogs that were classified as low eaters lost more proportional weight than dogs classified as medium to good eaters. Old dogs (8–13 years) lost more proportional weight than young dogs (1–4 years). At week 2, stray dogs had lost less proportional weight compared to relinquished dogs.

For the dogs for which BCS was evaluated both at intake and after 12 days (n = 46), BCS significantly decreased from day 1 (mean = 5.3) to day 13 (mean = 5.0, paired two-tailed t-test: semd = 0.32, 95% CI = 0.17–0.46, t[45] = 4.50, p < 0.001).

## Discussion

In this study, we investigate the usefulness of sensor-based assessment of nocturnal activity patterns, using accelerometers, as an additional measure of welfare in shelter dogs. We did this by studying the dynamics of nocturnal activity responses over time during the first 13 days in the shelter, while also evaluating physiological and behavioural measures of stress.

We found that sheltered dogs during the first three days in the shelter showed more nocturnal activity and higher UCCRs than matched control pet dogs in their own homes. Both the nocturnal activity and urinary cortisol/creatinine ratio (UCCR) of shelter dogs decreased over time from the first day(s) in the shelter to after 12 days in the shelter, but were then still higher than levels of pet dogs. Behavioural observations showed that sheltered dogs had higher 'active awake' behaviours during the first nights compared to later nights and spent more time on their resting place over days, and that behavioural indicators of stress also changed over the nights. These findings support that accelerometer measures are observations of interest as a lack of rest or sleep may reflect or lead to compromised welfare, whereas sufficient resting and sleeping can contribute to good welfare.

### Nocturnal activity of sheltered dogs as an indicator of adaptation

Under normal circumstances, pet dogs have about 3 sleep cycles of ~16–20 minutes each per hour at night [24] and spent ~96–98% of the night resting [13,26]. In the pet dog population in this study, similar results were found: accelerometer results showed that pet dogs rested 94% of the time during the night. However, our pet dogs had a lower number of bouts >15 minutes (on average 6.6 bouts per 4 hours) that were long enough to complete a sleep cycle, than reported in Adams & Johnson [24]. During the first night in the shelter, sleep characteristics of

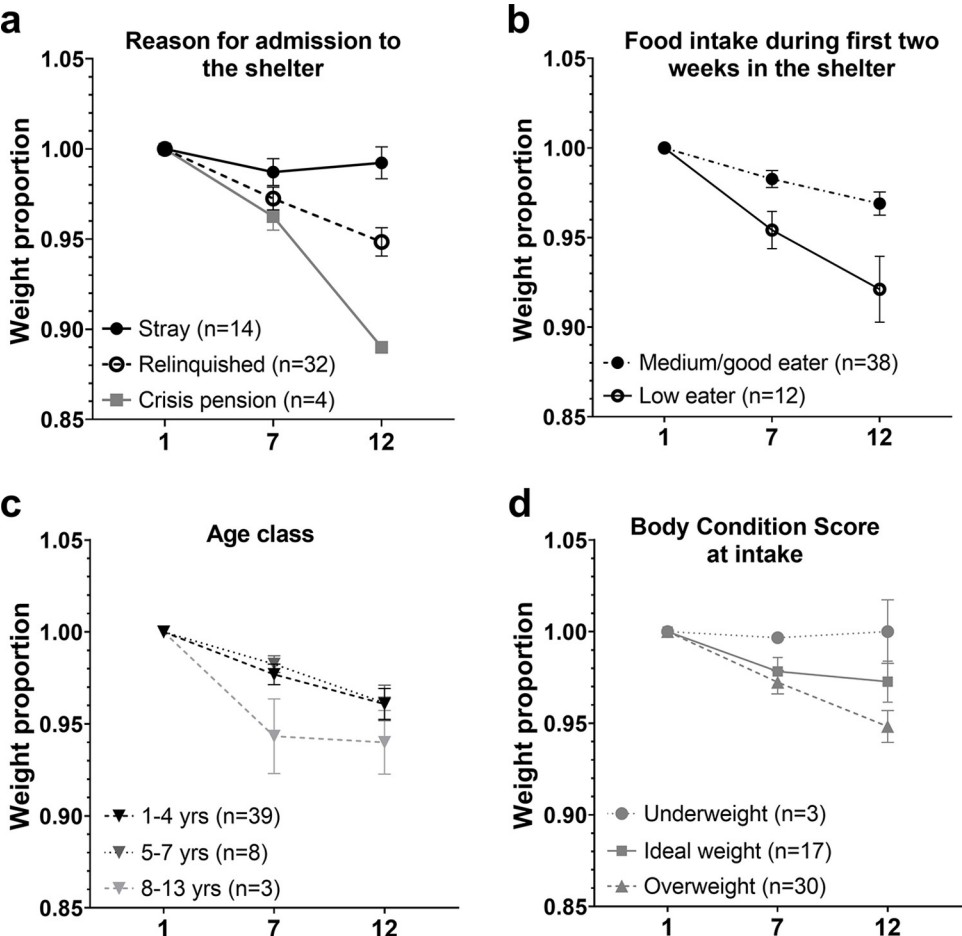

**Fig 9.** Shelter dogs results for weight proportional change over days, divided in a) reason for admission to the shelter, b) food intake during first two weeks in the shelter, c) age class and d) body condition score at intake in the shelter. Means and standard error of the mean (SEM) for the shelter dog group on day 1, 7 and 12 (x-axis) in the shelter. For graph d), no significant effect was found, although this interaction between BCS and day was dropped last from the model.

our shelter dogs as measured with the accelerometer were clearly aberrant from those of our pet dogs, with on average 72% of time spent in rest and on average only 3.4 times during the 4 hours an inactive bout was long enough to complete a sleep cycle. Other studies also found that dogs sleep less in shelters than in homes. For example, Owczarczak-Garstecka & Burman [22] found sheltered dogs to sleep 71.6% of the night-time and Hoffman et al. [26] found sheltered dogs to rest about 89% of the night-time. Our previous study, with another dog population but in the same shelter environment as in this study, found that shelter dogs rested 88% of the time during the first night [13]. Shelter environments can differ greatly, and it has yet to be studied what stressors contribute most to a stress response in dogs. For example, the effect of novelty versus specific stressors such as noises has yet to be studied.

Resting patterns as an indicator of adaptation to novel environments were already studied in other species [17–19] and in dogs [13]. However, the dynamics of this response were largely unknown. In this study, the highest decrease of nocturnal activity was visible during the first three nights in the shelter, but even after 12 nights, dogs were still longer active at night than dogs in the pet dog control group. Shelter dogs had higher interindividual variability in most

parameters, and especially during the first days in the shelter. This reflects a dynamic adaptive response which might not have been completed for all dogs yet after 12 days in the shelter, which highlights the importance to individually monitor each dog. Also, this activity response also takes generally longer than one night, suggesting a 'more-than-one-night-effect' instead of a 'first-night effect' in the shelter, supporting what we reported earlier [13]. These results are supported by our UCCR measurements, which follow the same pattern as the nocturnal activity parameters. This pattern of highest levels of nocturnal activity and cortisol responses during the first 1 to 3 days in a shelter has been described before in dogs in a different study, with high plasma cortisol levels on day 1–3, with a decline thereafter [3].

Some behavioural indicators of stress observed in this study changed over time in the shelter. Although oral behaviour, which was yawning and lip/snout licking combined, did not change over days in the shelter, shelter dogs overall showed less body shaking during the first night in the shelter and less dogs showed paw lifting over time in the shelter. In previous studies, paw-lifting decreased over time from day 1 to 10 after entering kennels [8,12]. Other studies previously found no difference between a home and kennel environment for paw lifting, lip licking, yawning and body shaking [9]. Body shaking is seen commonly in stressful situations, or mainly post-stressor as an expression of relief, by 'shake off'/coping with the tension [7,45,46]. Smaller dogs showed less autogrooming during the first nights in the shelter compared to larger dogs, with an increase over days. An increase in autogrooming over time in kennels has been seen before, together with a decrease in other stress related behaviours [8,12] and could therefore reflect a decrease in stress, maybe a recovery of self-care, and thus adaptation.

It is not known whether dogs can fully adapt to shelter situations, and how long dogs need to adapt, and individual differences are large. Dogs that had been in the shelter for more than 30 days seemed to have better welfare, as measured by a quality of life scoring tool, than newly admitted dogs, suggesting they adapted to a certain extent to the kennel environment [47]. However, even after being in stressful situations for years, dogs still had higher cortisol levels and behavioural responses than dogs in homes [37]. Also, even after 10 weeks in a training facility, UCCR levels of dogs did not return to levels measured (pre-facility) in their homes [8]. However, long-term elevated UCCR levels might be functional and even adaptive, when animals remain in stressful situations and may therefore need to respond faster to stressors. On the other hand, prologued exposure to stressors may also lead to dysregulation of the HPA axis. The latter has also been observed in dogs as they showed a decrease in plasma cortisol in an animal shelter but not in ACTH levels, indicating that the stress system can be reactivated swiftly when needed and the HPA axis might be longer activated than the cortisol response would suggest [10]. Overall, it is unknown whether (chronically) elevated cortisol levels imply a negative emotional state. More research is needed to study adaptability of dogs to stressors, including ACTH measures in blood, although invasive, to study the dynamics of the HPA axis response and potentially even response intensity [48]. In addition, behaviour of shelter dogs, such as activities and position in the kennel, can be quantified remotely and automated with 3D computer vision technology [49], and can therefore also be useful to monitor nocturnal activity.

## Accelerometers to measure sleep or rest

Accelerometry or observation of behaviour as used in this study cannot objectively measure sleep stages or sleep quality, which gives interpretation limitations. For example, in cows, lying posture did not accurately indicate sleep stages [50], and dogs are known to be able to sleep without having their eyelids fully closed. Non-invasive measurements with high specificity of sleep versus awake state and different sleep stages can only be determined with

polysomnographic recordings [51]. It is hypothesized that especially the absence of REM sleep might have an adaptive role in a novel or stressful environment, as awareness of animals is reduced during REM sleep posing a risk when being confronted with threatening stimuli [24]. Therefore, REM sleep is mainly seen in 'safe' environments, or REM sleep has a later onset during sleep in novel environments [52]. In this study, we were not able to use EEG recordings in our population of dogs and therefore to measure changes in REM sleep. More research with the help of REM sleep evaluating techniques such as EEG is needed to evaluate the effect of REM sleep in adaptation and therefore welfare of animals.

Although accelerometry brings insight into nocturnal (in)activity, accelerometer recordings are unable to identify 'quiet wakefulness', which is lying inactive but awake. Instead of active behaviour such as moving or standing, lying inactive but awake can also be an expression of continuous restlessness and alertness [53] which especially during the night alerts on adaptation problems and may impede welfare if proceeding on the long term. Although a problem with sleeping is a symptom of clinical depression in humans [54], studies in sheltered dogs show that 'quiet wakefulness' during the day was not associated with depression-like characteristics [55,56]. The underlying motivations and interpretation of 'quiet wakefulness' are still unclear. More research is needed into 'quiet wakefulness', with context interpretative studies, such as differences between the night and day, to evaluate the mechanisms and motivations for sound interpretations of this behaviour.

## Body weight loss

Sheltered dogs lost significant body weight after 1 and 2 weeks in the shelter compared to their body weight directly after intake at the shelter, which was described before [13]. Average body condition score (BCS) decreased from 5.3 to 5 during these two weeks in the present study. As makes sense, dogs that were classified as low eaters (23.6%) lost relatively more body weight than dogs classified as medium to good eaters. Older dogs lost more proportional weight compared to younger dogs. Relinquished dogs lost more proportional weight compared to strays. No difference in BCS at intake in strays versus relinquished dogs was found, and strays were not labelled different from relinquished dogs regarding food intake. This supports previous findings [13] and the hypothesis that strays may respond less strongly to entering shelters than dogs that were relinquished directly from homes [12].

Weight loss can be stress-related (in cats: [57,58]). However, many dogs in the general dog population are overweight (BCS = >5 out of 9 [59]) and therefore mainly these fatter dogs might lose weight after entering a shelter with a better balanced diet. In our study, 58.2% of shelter dogs had a BCS of >5 when entering the shelter, which is comparable to the overweight prevalence in dogs described in the literature which is around 53% [60] and 56% [61]. Hence, these intermingled factors makes the conclusion whether shelter weight losses are due to stress or due to a reduction of overweight dogs more difficult. However, no significant effect of BCS on body weight loss was found in the shelter dogs, although a trend showed that overweight dogs lost more relative weight (Fig 9D). Lastly, higher daytime activity levels can lead to more body weight loss. Daytime activity levels were not recorded during this study, therefore this relation was not controlled for. As a recommendation for future studies, studying this relation but also evaluating body weight and BCS over a longer time period can provide the additional information, whether dogs return to pre-shelter weight or to ideal body condition scores over time in the shelter.

## Factors impacting on nocturnal activity and cortisol responses

Several factors of influence on our findings of activity and cortisol responses are mentioned below.

**Body weight class.** In the shelter, lower body weight classes, i.e. smaller dogs, were more active and showed more restlessness during nights than heavier weight classes (as visible in the accelerometer reported *# inactive bouts*, *# inactive bouts >15 min*, and video-observed *% of active behaviour*, mainly *stationary behaviour*). Smaller dogs also had higher UCCR than larger dogs in the shelter as found in another study as well [62,63]. A previous study also showed that during controlled movements (i.e. on-leash), accelerometers detected activity differently for different body weight classes, with higher activity counts for smaller dogs [28]. However, smaller dogs also seem to show a higher cortisol response and activity levels when placed in a kennel environment, especially during the first days in a shelter [13,27], but not when data were averaged during the period from intake until dogs were ready for adoption, for shelter dogs over all nights [26]. This suggests that smaller dogs show a higher stress response compared to larger dogs, when being placed in a kennel environment. A higher stress response in smaller dogs when entering a kennel environment is supported by behavioural measures in this study, as smaller dogs showed less autogrooming during the first nights in the shelter compared to larger dogs. One possible explanation is that smaller breeds are trained less [64] and have less socialisation experiences during puppyhood (7–16 weeks old) [65] and might therefore be more prone to stress in a shelter environment. However, in this study we found no interaction effects between night in the shelter and body weight. A complicating factor for interpretation is that higher UCCRs in smaller dogs can also be explained by the relatively small muscle mass [63], as creatinine production is proportional to muscle mass [66]. Therefore, the effect of general higher cortisol and activity levels or activity detection by accelerometers in smaller dogs, versus the effect of a kennel environment needs to be explored in more detail in future studies. For example, by using within-subject comparisons in-shelter versus post-adoption or additional validation studies with controlled activity movements in small and larger dogs, activity differences for smaller versus larger dogs in homes versus animal shelters can be compared.

**Kennel history.** Shelter dogs that had a known history of staying in a kennel environment, were less active during the night, showed more body shaking and had lower UCCR levels than dogs that had not stayed in a kennel before. This effect was mainly visible during the first nights in the shelter and suggests a habituation process by former experiences. Previously, Rooney and colleagues [8] described a higher increase of UCCR in unhabituated Labradors when entering a training facility compared to habituated dogs, but contrary Part and colleagues [9] did not find an effect of history of kennelling on any stress parameter. Our results in this study suggest that prior experience with kennels might dampen a dog's response when entering an animal shelter, although reliable information about the history of our shelter dog population was relatively scarce. Habituation to a kennel environment as suggested by Rooney and colleagues [8,41], for example by bringing a dog to day care in kennels before submission to a shelter environment, might help dogs to adapt. However, to make a solid recommendation, prior habituation effects of kennels need to be further studied.

**Age class and sex.** Young dogs were more stationary (sitting/standing) and had a higher rate per minute of transitions in the different activity behaviours than middle-aged dogs, which suggests that the last group was more restless. However, we found no effect of age in our accelerometer measures or UCCR levels. Females in our study showed more active behaviour than males during the night, which adds to the literature describing bitches to show a higher behavioural response to acute stressors [14] and being more susceptible to environmental stress [67].

**Outlier details.** Interestingly, one of the shelter dogs, a French Bulldog, had a very high (outlier) number of recumbent bouts visible in the accelerometer data. Behavioural observations showed that this dog did not stay in a head down position for long and had therefore

more fragmented rest, although total activity did not deviate from other dogs. Disturbed sleep has been described in brachycephalic dogs before, likely due to breathing issues as part of brachycephalic obstructive airway syndrome [68]. However, this dog's number of recumbent bouts did not visibly differ over the days, therefore not likely explaining overall changes in disrupted nocturnal activity in the shelter dog group.

## Conclusion

Shelter dogs had disrupted nocturnal resting patterns and increased UCCR levels compared to pet dogs, especially during the first nights in the shelter. These activity and UCCR levels decreased but did not return to pet dog levels after 12 days in the shelter, which suggests partial adaptation to the shelter environment, but not total adaptation. Sensor-supported identification of nocturnal resting patterns, using accelerometers, can be a useful addition to welfare assessments in animal shelters as shown by paralleled physiological and behavioural parameter outcomes.

This study highlights the importance of evaluating individual dogs when transferring to a new environment, raises concerns about the amount of nocturnal rest in the shelter, and provides opportunities to consider improving nocturnal rest by more suitable housing and management. In future studies, the effect of different properties of stressors in a shelter environment versus the novelty of the shelter environment can be studied by further evaluating responses of dogs after a change in environment, e.g. in shelter versus novel (re)homes as a control.

## Supporting information

**S1 Table. Demographics per shelter dog (SD) and control pet dog (CPD).** Estimated breed group [31], age class (in years, [13]), sex (female = f, male = m), neuter status (yes = y, no = n, unknown =?), reason for admission to the shelter (relinquished = R, stray = S, crisis boarding = CB), body weight class (in kg), body condition score (BCS) at intake in shelter (underweight BCS 1-3/ideal weight BCS 4-5/overweight BCS 6-9), food intake label (low eater or medium/good eater during the first two weeks in the shelter) and nocturnal video observations available (yes = y, no = n).
(DOCX)

**S2 Table. Full model results of nocturnal activity accelerometer outputs: Vector Magnitude Counts per minute (VMCpm) in the shelter dog group.** Estimated parameter (EP) and 95% confidence intervals (CI) of *VMCpm* during the night (0:00–4:00 h) for night (after intake) and neuter status, that both significantly explained the *VMCpm* variability. Conditional F-testing revealed F, DF's and significance of the different terms in the models. [1] Estimated mean on reference night and neuter status. [2] Estimated ratio of mean of specified night and mean on reference night. [3] Estimated ratio of mean of specified neuter status and mean in reference neuter status.
(DOCX)

**S3 Table. Full model results of nocturnal activity accelerometer outputs: Percentage of time spent active in the shelter dog group.** Estimated parameter (EP) and 95% confidence intervals (CI) of *% active* during the night (0:00–4:00 h) for night (after intake) and neuter status, that both significantly explained the *% active* variability. Conditional F-testing revealed F, DF's and significance of the different terms in the models. [1] Estimated mean on reference night and neuter status. [2] Estimated ratio of mean of specified night and mean on reference

night. [3] Estimated ratio of mean of specified neuter status and mean of reference neuter status.
(DOCX)

**S4 Table. Full model results of nocturnal activity accelerometer outputs: Number of inactive bouts in the shelter dog group.** Estimated parameter (EP) and 95% confidence intervals (CI) of *# inactive* during the night (0:00–4:00 h) for night (after intake) and other factors that significantly explained the *# inactive* variability. Conditional F-testing revealed F, DF's and significance of the different terms in the models. [1] Estimated mean on reference night, weight class and kennel history. [2] Estimated ratio of mean of specified night and mean on reference night. [3] Estimated ratio of mean of specified weight class and mean of reference weight class. [4] Estimated ratio of mean of specified kennel history and mean of reference kennel history.
(DOCX)

**S5 Table. Full model results of nocturnal activity accelerometer outputs: Number of inactive bouts >15 minutes in the shelter dog group.** Estimated parameter (EP) and 95% confidence intervals (CI) of *# inactive >15 min* during the night (0:00–4:00 h) for night (after intake) and other factors that significantly explained the *# inactive >15 min* variability. Conditional F-testing revealed F, DF's and significance of the different terms in the models. [1] Estimated mean on reference night and weight class. [2] Estimated difference between mean of specified night and mean on reference night. [3] Estimated difference between mean of specified weight class and mean of reference weight class.
(DOCX)

**S6 Table. Model results for urinary cortisol/creatinine ratio (UCCR) of the shelter dog group.** Estimated parameter (EP) and 95% confidence intervals (CI) of *UCCR* for day (after intake) and other factors that significantly explained *UCCR* variability. Conditional F-testing revealed F, DF's and significance of the different terms in the models. [1] Estimated mean on reference day, weight class and kennel history. [2] Estimated ratio of mean of specified day and mean on reference day. [3] Estimated ratio of mean of specified weight class and mean of reference weight class. [4] Estimated ratio of mean of specified kennel history and mean of reference kennel history.
(DOCX)

**S7 Table. Model results for nocturnal activity behaviour: Percentage of time showing active behaviour in the shelter dog group.** Estimated parameter (EP) and 95% confidence intervals (CI) of *% active behaviour* during the night (0:00–4:00 h) for night (after intake) and other factors that significantly explained *% active behaviour* variability. Conditional F-testing revealed F, DF's and significance of the different terms in the models. [1] Estimated mean on reference night, weight class, sex and kennel history. [2] Estimated ratio of mean of specified night and mean on reference night. [3] Estimated ratio of mean of specified weight class and mean of reference weight class. [4] Estimated ratio of mean of specified sex and mean of reference sex. [5] Estimated ratio of mean of specified kennel history and mean of reference kennel history.
(DOCX)

**S8 Table. Model results for nocturnal activity behaviour: Percentage of time showing recumbent head up in the shelter dog group.** Estimated parameter (EP) and 95% confidence intervals (CI) of *% recumbent head up* during the night (0:00–4:00 h) for night (after intake) and other factors that significantly explained *% recumbent head up* variability. Conditional F-testing revealed F, DF's and significance of the different terms in the models. [1] Estimated mean on reference night, weight class and kennel history. [2] Estimated ratio of mean of specified night and mean on reference night. [3] Estimated ratio of mean of specified weight class and

mean of reference weight class. [4] Estimated ratio of mean of specified kennel history and mean of reference kennel history.
(DOCX)

**S9 Table. Model results for nocturnal activity behaviour: Percentage of time showing stationary behaviour in the shelter dog group.** Estimated parameter values (EP) and 95% confidence intervals (CI) of % *stationary* during the night (0:00–4:00 h) for night (after intake) and other factors that significantly explained % *stationary* variability. Conditional F-testing revealed F, DF's and significance of factors in the model. [1] Estimated mean in reference night, weight class, age class and kennel history. [2] Estimated ratio of mean of specified night and mean on reference night. [3] Estimated ratio of mean of specified weight class and mean in reference weight class. [4] Estimated ratio of mean of specified age class and mean of reference age class. [5] Estimated ratio of mean of specified kennel history and mean of reference kennel history at the same night.
(DOCX)

**S10 Table. Model results for nocturnal activity behaviour: Percentage of time showing movement behaviour in the shelter dog group.** Estimated parameter values (EP) and 95% confidence intervals (CI) of % *of movement* during the night (0:00–4:00 h) for night (after intake) and neuter status, that both significantly explained the % *of movement* variability. Conditional F-testing revealed F, DF's and significance of factors in the model. [1] Estimated mean on reference night and neuter status. [2] Estimated ratio of mean of specified night and mean on reference night. [3] Estimated ratio of mean of specified neuter status and mean of reference neuter status.
(DOCX)

**S11 Table. Model results for nocturnal activity behaviour: Rate per minute (RPM) of transitions in the different activity behaviours in the shelter dog group.** Estimated parameter values (EP) and 95% confidence intervals (CI) of *transition in movements (rpm)* during the night (0:00–4:00 h) for night (after intake) and age class, that both significantly explained the *transitions in movement (rpm)* variability. Conditional F-testing revealed F, DF's and significance of factors in the model. [1] Estimated mean on reference night and age class. [2] Estimated ratio of mean of specified night and mean on reference night. [3] Estimated ratio of mean of specified age class and mean of reference age class.
(DOCX)

**S12 Table. Model results for nocturnal behavioural indicators of stress: Rate per minute (RPM) of oral behaviour (yawning + lip/snout licking) in the shelter dog group.** Estimated parameter values (EP) and 95% confidence intervals (CI) of *oral behaviour (RPM)* when the dog was in sight of the camera and active, during the night (0:00–4:00 h) for night (after intake). Conditional F-testing revealed F, DF's and significance of factors in the model. [1] Estimated mean on reference night. [2] Estimated ratio of mean of specified night and mean on reference night.
(DOCX)

**S13 Table. Model results for nocturnal behavioural indicators of stress: Proportion of time spent autogrooming in the shelter dog group.** Estimated parameter values (EP) and 95% confidence intervals (CI) of *autogrooming (proportion of time)* when the dog was in sight of the camera and active, during the night (0:00–4:00 h) for night (after intake) and other factors that significantly explained *autogrooming* variability. Conditional F-testing revealed F, DF's and significance of factors in the model. [1] Estimated mean in reference night, weight class and age

class. [2] Estimated ratio of mean of specified night and mean on reference night. [3] Estimated ratio of mean of specified weight class and mean in reference weight class at the same night. [4] Estimated ratio of mean of specified age class and mean of reference age class at the same night.
(DOCX)

**S14 Table. Model results for nocturnal behavioural indicators of stress: Rate per minute (RPM) of body shaking in the shelter dog group.** Estimated parameter values (EP) and 95% confidence intervals (CI) of *body shaking (RPM)* when the dog was in sight of the camera and active, during the night (0:00–4:00 h) for night (after intake) and other factors that significantly explained *body shaking* variability. Conditional F-testing revealed F, DF's and significance of factors in the model. [1] Estimated mean in reference night, weight class and age class. [2] Estimated ratio of mean of specified night and mean on reference night. [3] Estimated ratio of mean of specified weight class and mean in reference weight class. [4] Estimated ratio of mean of specified kennel history and mean of reference kennel history.
(DOCX)

## Acknowledgments

The authors would like to thank the employees of Animal Shelter DOA in Amsterdam for their help with the data collection, and all owners that participated with their pet dog. We would also like to thank the students that greatly helped to collect data: Chantal Houben, Katja van den Brink, Laura Rossetti and Nikki Delis.

## Author Contributions

**Conceptualization:** Janneke E. van der Laan, Claudia M. Vinke.

**Data curation:** Janneke E. van der Laan.

**Formal analysis:** Janneke E. van der Laan.

**Funding acquisition:** Janneke E. van der Laan, Claudia M. Vinke, Saskia S. Arndt.

**Investigation:** Janneke E. van der Laan.

**Methodology:** Janneke E. van der Laan, Claudia M. Vinke.

**Project administration:** Janneke E. van der Laan.

**Resources:** Janneke E. van der Laan.

**Software:** Janneke E. van der Laan.

**Supervision:** Claudia M. Vinke, Saskia S. Arndt.

**Validation:** Janneke E. van der Laan.

**Visualization:** Janneke E. van der Laan.

**Writing – original draft:** Janneke E. van der Laan.

**Writing – review & editing:** Janneke E. van der Laan, Claudia M. Vinke, Saskia S. Arndt.

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
