## [Decision Letter · Decision Letter 0]

4 Apr 2023

PONE-D-23-07468Sensor-supported measurement of adaptability of dogs (Canis familiaris) to a shelter environment: Nocturnal activity and behaviourPLOS ONE

Dear Dr. van der Laan,

Thank you for submitting your manuscript to PLOS ONE. After careful consideration, we feel that it has merit but does not fully meet PLOS ONE’s publication criteria as it currently stands. Therefore, we invite you to submit a revised version of the manuscript that addresses the points raised during the review process.

We look forward to receiving your revised manuscript.

Kind regards,

Muhammad Mazhar Ayaz, Ph.D

Academic Editor

PLOS ONE

Journal Requirements:

2.  You indicated that ethical approval was not necessary for your study. We understand that the framework for ethical oversight requirements for studies of this type may differ depending on the setting and we would appreciate some further clarification regarding your research. Could you please provide further details on why your study is exempt from the need for approval and confirmation from your institutional review board or research ethics committee (e.g., in the form of a letter or email correspondence) that ethics review was not necessary for this study? Please include a copy of the correspondence as an ""Other"" file.

Reviewers' comments:

Reviewer's Responses to Questions

**Comments to the Author**

1. Is the manuscript technically sound, and do the data support the conclusions?

Reviewer #1: Yes

2. Has the statistical analysis been performed appropriately and rigorously? 

Reviewer #1: Yes

3. Have the authors made all data underlying the findings in their manuscript fully available?

Reviewer #1: Yes

4. Is the manuscript presented in an intelligible fashion and written in standard English?

Reviewer #1: Yes

5. Review Comments to the Author

Reviewer #1: Thanks for submitting this paper which is a good additional to the literature on the effect of shelters of canine welfare. The paper is comprehensive and all methods and results fully presented. The conclusions seem to be reasonable.

DETAILED Comments

There are a number of English errors that will need to be corrected, e.g. Line 59 (may be), 108 (returned to their owners), the use of the word 'epoch' is unusual, 650 (admitted), 658 (indicating), 675 (safe), 705 (better balanced diet), 727 (data were) and 750 (dampen).

I would like to see a better explanation why some results were excluded (e.g. 286-7)

Line 339 - is this supposed to be after 13 days?

Line 733 - please explain what you mean by socialised less as used here.

Line 765 - the word 'before' is repeated.

6. PLOS authors have the option to publish the peer review history of their article (what does this mean?). If published, this will include your full peer review and any attached files.

Reviewer #1: **Yes: **Mandy B A Paterson

---

## [Author Response · Author response to Decision Letter 0]

6 May 2023

Points raised by the reviewer(s):

Reviewer #1: Thanks for submitting this paper which is a good additional to the literature on the effect of shelters of canine welfare. The paper is comprehensive and all methods and results fully presented. The conclusions seem to be reasonable.

DETAILED Comments

There are a number of English errors that will need to be corrected, e.g. Line 59 (may be), 108 (returned to their owners), the use of the word 'epoch' is unusual, 650 (admitted), 658 (indicating), 675 (safe), 705 (better balanced diet), 727 (data were) and 750 (dampen). Thank you for pointing out these errors. We have corrected them, and explained what we meant with ‘epoch’. 

I would like to see a better explanation why some results were excluded (e.g. 286-7) These data were excluded as these behaviours were observed too little to perform (mixed model) analysis. We chose for the 10% cutoff conform Hiby et al. (2006) and Part et al. (2014).

Line 339 - is this supposed to be after 13 days? Yes, thank you, corrected.

Line 733 - please explain what you mean by socialised less as used here. We changed the sentence into: One possible explanation is that smaller breeds are trained less [64] and have less socialisation experiences during puppyhood (7-16 weeks old) [65].

Line 765 - the word 'before' is repeated. We corrected this error.

---

## [Editor Report · Decision Letter 1]

16 May 2023

Sensor-supported measurement of adaptability of dogs (Canis familiaris) to a shelter environment: Nocturnal activity and behaviour

PONE-D-23-07468R1

Dear Dr. van der Laan,

We’re pleased to inform you that your manuscript has been judged scientifically suitable for publication and will be formally accepted for publication once it meets all outstanding technical requirements.

Kind regards,

Muhammad Mazhar Ayaz, Ph.D

Academic Editor

PLOS ONE
---

## [Editor Report · Acceptance letter]

7 Jun 2023

PONE-D-23-07468R1 

Sensor-supported measurement of adaptability of dogs (*Canis familiaris*) to a shelter environment: Nocturnal activity and behaviour 

Dear Dr. van der Laan:

I'm pleased to inform you that your manuscript has been deemed suitable for publication in PLOS ONE. Congratulations! Your manuscript is now with our production department. 

Kind regards, 

on behalf of

Dr. Muhammad Mazhar Ayaz 

Academic Editor

PLOS ONE